# Tumor-Derived Membrane Vesicles: A Promising Tool for Personalized Immunotherapy

**DOI:** 10.3390/ph15070876

**Published:** 2022-07-16

**Authors:** Jiabin Xu, Wenqiang Cao, Penglai Wang, Hong Liu

**Affiliations:** 1School of Stomatology, Xuzhou Medical University, Xuzhou 221004, China; jabbyxu@foxmail.com (J.X.); wpl0771@163.com (P.W.); 2Affiliated Stomatological Hospital of Xuzhou Medical University, Xuzhou 221004, China; 3Zhuhai Jinan Selenium Source Nanotechnology Co., Ltd., Jinan University, Zhuhai 519000, China; sesource_cwq@163.com

**Keywords:** tumor-derived membrane vesicles, cancer therapy, tumor vaccine, personalized immunotherapy

## Abstract

Tumor-derived membrane vesicles (TDMVs) are non-invasive, chemotactic, easily obtained characteristics and contain various tumor-borne substances, such as nucleic acid and proteins. The unique properties of tumor cells and membranes make them widely used in drug loading, membrane fusion and vaccines. In particular, personalized vectors prepared using the editable properties of cells can help in the design of personalized vaccines. This review focuses on recent research on TDMV technology and its application in personalized immunotherapy. We elucidate the strengths and challenges of TDMVs to promote their application from theory to clinical practice.

## 1. Introduction

Cancer is the leading cause of human death globally and a significant disease reducing life expectancy. According to the International Agency for Research on Cancer (IARC), 19.3 million new cases and millions of cancer deaths were estimated worldwide in 2020 [1]. Clinically, the primary treatments for cancers are surgical resection, radiotherapy and chemotherapy, but these methods may cause tumor recurrence, damage to normal tissues, and toxic side effects caused by a lack of precise targeting [2]. In contrast to conventional treatments, tumor immunotherapy, which refers to the regulation of the patient’s immune system to fight against tumors, has become the mainstream of research and clinical practice. Immunotherapy by activating immune cells to trigger a systemic anti-tumor immune response can eradicate primary and distant tumors, establishing long-term immune memory to prevent tumor recurrence [3]. Tumor immunotherapy has now yielded exciting results in hematological cancers, lymphoma and myeloma [4]. Monoclonal antibodies, immune checkpoint therapy (ICT) and chimeric antigen receptor (CAR) T cell therapy have achieved significant clinical efficacy [5,6]. However, there remain many challenges to the widespread clinical application of immunotherapy. Monoclonal antibodies can exhibit off-target toxicity and adverse immunological effects. Only 13% of patients showed a significant immune response to ICT [7]. Effective CAR T-cell therapy for hematological cancers has not significantly impacted the treatment of more prevalent solid epithelial cancers [8]. Combined immunotherapy has better therapeutic effects while at the cost of more acute inflammatory side effects [9,10]. Therefore, how to safely and effectively drive the immune response against cancer remains an urgent issue for tumor immunotherapy.

The gap between the tumor microenvironment in the body is the main reason for the significant gap in the efficacy of tumor immunotherapy. In particular, the immunosuppressive microenvironment and high interstitial pressure in solid tumors make it difficult for drugs to penetrate and act inside the tumor, thus contributing to the immune escape of tumor cells and making treatment very difficult. Tumors can be divided into three types [11]: the first is the immune desert type, which lacks tumor-infiltrating lymphocytes (TILs) in the central and peripheral regions. PD-1/PD-L1 therapy is clinically ineffective for this type. The second is the immune-excluded type, which has many TILs at the tumor edge but forms an immune desert in the central area. After treatment with anti-PD-L1/PD-1, stroma-associated T cells can show signs of activation and proliferation without deeper infiltration. The third, the immunoinflammatory type, has TILs in the central and peripheral regions and contains effectors such as monocytes and pro-inflammatory factors around the tumor.

Nevertheless, tumor escape suppresses the immune response. Tumor mutational burden (TMB) due to intratumoral heterogeneity also has implications for treatment. As TMB increases, more neoantigens are released, triggering more robust T cell responses for better therapeutic effects [12,13]. The amount and type of gut microbiota affect the incidence of cancer and the body’s susceptibility to treatment. The study showed that ICT had no significant effect on tumor-bearing mice lacking gut microbes [14,15,16]. The higher the abundance of probiotics and the more CD8^+^ T and CD4^+^ T cells in the surrounding blood, the better effect of anti-PD-1 on melanoma.

Fortunately, the development of nanomaterials has opened up more options for tumor immunotherapy. Protein particles, vesicles, liposomes, micelles, inorganic particles and metal-organic frameworks (MOFs) are used to enhance tumor immunogenicity and inhibit tumor growth [17,18,19]. However, the foreign substances and the carriers involved in drug transport are also easy to be eliminated by the immune system [20]. Polyethylene glycol (PEG) is commonly used to modify the surface of nanoparticles. It can protect the nanoparticles from clearance by the immune system and prolong drug half-life. However, recent studies found that PEG can accelerate the blood clearance effect, reducing the drug’s efficacy [21]. The major limitation to the broader success of immunotherapy treatments is not the lack of rational therapeutic targets but rather how to successfully reach these targets at the right time and place. Therefore, it is necessary to develop personalized therapeutic modalities that promote systemic and durable anti-tumor immunity to eradicate malignancies and prevent metastasis and recurrence completely [22].

Cell-derived membrane vesicles carry abundant recognition units that endow them with high biological specificity. Membrane vesicle camouflage can bypass clearance by immune cells, has a longer circulation time and upon reaching the tumor site, its bilayer lipid structure can fuse directly with tumor cells to utilize the treatment [2,23,24]. Membrane vesicles can improve bioavailability, effectively target treatment sites, reduce drug side effects and promote drug retention and sustained release in tissues [25,26,27]. In addition, several clinical results suggest that membrane vesicles of their own origin are virtually immunogenic and non-toxic, while synthetic materials still have unavoidable safety concerns in terms of immunogenicity and toxicity [28]. A variety of cell-derived membrane vesicle nanoparticles have been studied, including erythrocytes, stem cells, dendritic cells, natural killer cells, fibroblasts and tumor cells [29,30]. In particular, tumor cells can expand indefinitely and can isolate membrane vesicles in large batches of cultures in vitro. Unlike other membrane vesicles, tumor-derived membrane vesicles have properties that are highly similar to those of homologous tumors. TDMVs can be divided into tumor cell membrane vesicles (TCMVs) and tumor extracellular vesicles (TEVs). TCMVs carry a comprehensive and complete set of proteins and surface antigens that can stimulate the body’s immune response and have great potential as cancer vaccines [25,31]. TEVs are significant players in intercellular signaling and information exchange, regulating tumor progression by promoting tumor invasion, extracellular matrix remodeling, angiogenesis and immunosuppression [23,32].

Personalized therapies have emerged in response to differences in the efficacy of immunotherapy for different individuals. Delivering personalized drugs to patients based on genomic alterations, specific biomolecules and biomarkers can ensure maximum therapeutic efficacy and minimal side effects, eliminating the time wastage caused by ineffective drugs. Due to their excellent homotargeting capabilities, TDMVs can solve the challenge of precise targeting of personalized therapies and overcome the complex and cumbersome synthesis process of stimulus-responsive nanomedicines [33]. An example of personalized nanomedicine based on cancer cell membranes was obtained using B16F10 and red blood cells (RBC) membrane fusion vesicles and coating hollow copper sulfide nanoparticles loaded with doxorubicin (DOX) [34]. This study showed that the nanoparticle system had significant homotypic targeting with increasing cycle time and could effectively kill tumors.

For tumor therapy, tumor cell-derived membrane vesicles have promising application prospects and unique advantages (Figure 1). This review introduces the use of tumor cell-derived membrane vesicles, including cell membranes (TCMVs), exosomes (TEXs), microvesicles (TMVs) and apoptotic bodies in immunotherapy, discussing their benefits and limitations. Finally, a brief summary of the current challenges and prospects for using cancer cell-derived membrane vesicles to improve immunotherapy is presented.

## 2. Tumor Cell-Derived Membrane Vesicles

### 2.1. Tumor Cell Membrane Vesicles

Immunotherapy not only kills tumors in situ but also inhibits tumor metastasis and recurrence. However, how to precisely target tumor cells across the tumor extracellular matrix has been the main reason limiting the further development of immunotherapy. Due to the presence of large amounts of collagen, the extracellular matrix (ECM) of tumors is denser than normal cells, and the dense ECM makes it difficult for drugs to reach the core of tumors. The ECM can also bind to many drugs, further reducing the therapeutic effect [35,36]. At the same time, the rapid proliferation of the tumor makes the lymphatic reflux function absent, and the hydraulic pressure in the interstitial space of tumor tissue increases, making it difficult for the drugs injected intravenously to penetrate the tumor to achieve the therapeutic effect [37,38].

Tumor cells have the ability to self-target and adhere to homologous tumor tissue, known as “homologous adhesion” of tumor cells [39,40,41]. Many studies have shown that the abnormal expression of specific antigens on the surface of tumor cells (galactose lectin-3, N-calmucin, immunoglobulin-like cell-adhesion molecule) and cell adhesion molecules (epithelial cell adhesion molecules) contribute to this adhesion effect [41,42,43]. In particular, the immunoglobulin-like cell-adhesion molecule (Ig-CAM), which is overexpressed on the surface of tumor cells, supports homologous adhesion through homologous interactions with each of the other proteins (e.g., calmodulin, integrins) on the surface of adjacent cells [44,45,46]. This homologous adhesion confers specific self-recognition and self-targeting capabilities to tumor cells, which will provide a precise self-targeting function for immunotherapy.

TCMVs are tumor cell membranes extracted by squeezing using ultrasound and filter head to form nanoscale bilayer lipid structured vesicles that do not carry tumor cell genetic material. TCMVs have the properties of immune escape, unlimited growth capacity, anti-cell death, prolonged circulation time and homologous targeting. These properties allow TCMVs-based drug delivery systems to be highly aggregated at tumor sites (Table 1). TCMVs encapsulate adriamycin or paclitaxel and target tumor sites by exploiting homologous targeting of tumor cells [47,48]. Wan et al. delivered DOX and sorafenib (Sfn) with TMCVs to modulate the tumor microenvironment to sensitize the immune response against tumor immunogenic cell death (ICD). TMCVs delivery vehicle shows high stability, biocompatibility and excellent anticancer properties [49,50]. Cell membrane vesicle surface proteins provide antigen, recruit and activate antigen-presenting cells (APCs), while the wrapped drug can be released to kill tumor cells. MCF-7 cell membrane-encapsulated indocyanine green (ICG) nanoparticles were used in a diagnostic and therapeutic platform [41]. Thanks to the homologous targeting of cancer cell membranes, the particles aggregated heavily at tumor sites and reduced aggregation in the liver and kidney, showing good photothermal response and excellent imaging properties. Rao et al. designed a nanoprobe of MDA-MB-435 cancer cell membrane-encapsulated upconversion nanoparticles [51]. Lu et al. devised a highly sensitive multicellular miRNA imaging strategy for the first time by simultaneously encapsulating Au nanoparticles and DSN modified with miRNA fluorescent detection probes into tumor cell capsules to achieve ultrasensitive multiplex miRNA imaging in living cells [52]. Zhang et al. overexpressed PD-1 on cancer cell membranes to enhance anti-tumor responses by disrupting the PD-1/PD-L1 immunosuppressive axis [53]. Jiang et al. expressed CD80 on the surface of tumor cell membranes to make them into antigen-presentable particles [54]. The particles could directly stimulate activated tumor antigen-specific T cells by binding to the T cell receptor and CD28, and activated T cells suppress tumor growth by killing tumor cells expressing the same antigen. Zhu et al. designed a viral mimetic ternary nanoengineering constructed from DNA, Gd (III) and tumor cell membranes to enable host-specific gene therapy [54]. Membrane fusion technology expands the application of TCMVs. The particles obtained by fusing erythrocyte membranes with TCMVs have the biocompatibility of erythrocytes and the precise targeting ability of tumor cells and are loaded with photosensitizers to obtain excellent anti-tumor effects [34,55,56]. TCMVs carry all the antigenic proteins of tumor cells, but their immunogenicity is low. Researchers have fused TCMVs with biological membranes such as bacterial membranes, dendritic cell membranes and macrophage membranes to improve the immunogenicity of the particles and enhance antigen delivery, obtaining better immune effects [2,57,58,59].

### 2.2. Tumor Extracellular Vesicles

Extracellular vesicles (EVs) are nano- or micron-sized particles composed of phospholipid bilayers constitutively or inducibly released by all cells, including exosomes, microvesicles and apoptotic bodies (Table 2). The components of natural extracellular vesicles mainly contain proteins, nucleic acids, lipids and metabolites, which act as message transmitters, regulate recipient cells’ physiological and pathological states by delivering signaling molecules to other cells, and participate in the development and progression of many diseases [82,83,84]. They reflect the metabolic state of the organism and the function of cells under different pathological conditions and have potential clinical diagnostic value. Meanwhile, extracellular vesicles are the organism’s components with low immunogenicity and good biocompatibility, exhibiting tissue targeting mediated by surface molecules (integral proteins and glycan) and the ability to transport biomolecules to recipient cells promising for drug delivery applications [85]. Currently, therapeutic drugs are loaded into extracellular vesicles by incubation, extrusion, electroporation, ultrasound and genetic modification for therapeutic research in diseases such as cardiac repair, liver diseases, lung diseases and tumors [86,87,88].

TEVs secreted by tumor cells play an essential role in tumor development. TEVs interacting with ECM can induce tumor cell adhesion and targeted migration [95]. On the other hand, TEVs carry a variety of bioactive substances (proteins, RNA, lipids) that enhance vascular permeability and create a microenvironment conducive to tumor cell metastasis [96,97,98]. For example, lung cancer cell exosome miRNA-9 promotes endothelial cell migration and angiogenesis through the downregulation of the SOCS5-JAK-STAT pathway [99]. Glioma or breast cancer cells secrete glutaminyl transferase-containing microvesicles that can alter the tumor microenvironment and promote tumorigenesis [100]. Tumor cells secrete DNA-containing apoptotic vesicles capable of transferring from source cells to normal cells, triggering the expression of oncogenes in fibroblasts [101]. In addition, EVs from tumor cells such as melanoma, breast and lung cancer have been reported to carry the immunosuppressive molecule PD-L1, which promotes tumor growth by misfiring T cells through the binding of the PD-L1 structural domain to PD-1 [102].

TEVs can be used not only as biomarkers or drug delivery vehicles due to their unique advantages but also for tumor immunotherapy. Many clinical trials evaluate the composition of TEVs as biomarkers for determining cancer staging and the effectiveness of oncology drug therapy (Table 3). Researchers are evaluating the expression profiles of exosomal miRNA and PD-L1 before and after immunotherapy in patients with non-small cell lung cancer and exploring the potential of exosomes as biomarkers for predicting the efficacy of anticancer drugs (NCT04427475). Koh et al. designed an extracellular vesicle-based immune checkpoint inhibitor that achieved tumor regression by blocking CD47 with SIRPα on phagocytes, leading to more extensive CD8^+^ T cell infiltration in tumors [103]. TEVs carry tumor-associated antigens (TAA) that stimulate in vivo through the Fas/FasL signaling pathway CD8^+^ T cells to kill tumors. Meanwhile, TEVs enhance antigen uptake by DCs and trigger more robust immune responses by inducing specific CD4^+^ T cell proliferation. TEVs play an important role in regulating immune cells in the tumor microenvironment as mediators between tumor and immune cells.

#### 2.2.1. Exosomes

Compared to synthetic nanoparticles, exosomes have the natural advantage of less toxicity and less rejection by the immune system. In addition, it allows them to cross some barriers, such as the placental barrier and blood–brain barrier [113,114,115]. Exosomes are not formed by direct outgrowth or shedding from the plasma membrane; instead, they are formed by inward outgrowth from the inner membrane, resulting in the formation of intracellular multi-vesicular vesicles, which then fuse with the plasma membrane and release exosomes into the extracellular compartment [89,90]. Exosomes are common membrane-bound nanovesicles that contain a variety of biomolecules such as lipids, proteins and nucleic acids. Exosome surface-bound proteins are derived from the cytoplasmic membrane from which they originate, and thus exosomes released from antigen-presenting cells, DCs and tumor cells have promising applications in vaccine development [116]. In addition, exosomes have characteristics such as lipid membranes and particle sizes in the nanoscale size, which can reduce the clearance or damage of their contents by complement or macrophages, thus prolonging the circulating half-life and improving biological activity. Chemical or biological modification of exosomes can enhance the potential therapeutic capacity of exosomes. Thus, exosomes can be used as drug delivery vehicles to treat diseases [117]. Exosomes have advantages such as good biocompatibility, low immunogenicity and the ability to cross the blood–brain barrier, which facilitate the delivery of nucleic acids or drugs. Exosomes carry tumor necrosis factor-related apoptosis-inducing ligands that transduce pro-apoptotic signals to different tumor cells, thereby inducing apoptosis in cancer cells and ultimately inhibiting tumor progression [118]. In addition, exosomes can transport small molecule compound drugs, such as paclitaxel and adriamycin, across the blood–brain barrier for targeted delivery [114,119]. Exosomes can eliminate the drug resistance properties of tumor cells, such as wrapping chemotherapeutic drugs into tumor cell-derived exosomes. They are preferentially internalized by tumor cells and subsequently release anti-tumor drugs, thereby reversing tumor cell resistance in vitro. There are differences in the surface protein composition of exosomes of different cellular origins, and their ability to transport RNA-like drugs varies. Mesenchymal cell-derived exosomes expressing the CD47 protein that protects cells from phagocytosis enhanced the ability to deliver miRNAs to pancreatic tumors, significantly improving overall survival in a mouse model [120].

The secretion of tumor-derived exosomes (TEXs) is mainly attributed to the overexpression of Rab3D and the acidic tumor microenvironment [121], which carries information characteristic of malignant tumors, such as NKG2-D-ligand on the surface of melanoma-derived exosomes [122]. It shows that TEXs play a crucial role in cancer metastasis and proliferation [123]. However, TEXs mode of communication with cells is still not fully understood. Current scholarship suggests that TEXs rely mainly on lipid rafts in the cytoplasmic membrane of target cells [124], cholesterol homeostasis [125] or membrane fusion [121] to enter the recipient cells. Uptake of TEXs by ordinary endothelial cells activates angiogenic signaling pathways in the cells to stimulate new blood vessel formation and promote tumor growth [126]. At the same time, TEXs can be secreted into the tumor microenvironment via autocrine and paracrine signaling to initiate epithelial-mesenchymal transition (EMT), which has a high potential to form tumor metastases once it spreads to the systemic circulation [127]. In addition, TEXs can evade immune surveillance and promote irrepressible metastatic progression [128]. TEVs can influence tumor progression by affecting the role of immune cells. It found that TEVs carry NKG2D ligands that bind directly to natural killer (NK) cells, downregulate NKG2D expression, inhibit NK cell activation and significantly reduce the recognition and killing effect of NK cells on tumors [129].TEVs can regulate macrophage polarization, organize phagocytosis of tumor cells by macrophages and enhance tumor cell drug resistance [130,131]. TEVs act on CD8^+^ T and CD4^+^ T cells through PD-L1 on the membrane surface, resulting in suppression of T cell function, a significant reduction in secreted effector cytokines (TNF-α, IL-2, TNF-α), and a significant reduction in the expression of CD69 and CD25, markers of CD8^+^ T cell and CD4^+^ T cell activation [132,133,134].

Engineered exosomes enhance anti-tumor capacity through precise targeting, high bioavailability and enhanced efficiency. Surface modification of TEXs through cellular transgene expression, chemical modification and electrostatic interactions can further enhance the targeting efficiency of exosomes to cancer cells. TEVs have been used as delivery systems for small molecule drugs, proteins and nucleic acids [135,136]. In addition, TEXs promote resistance to various chemotherapeutic agents and antibodies, which provides a multi-component diagnostic window for tumor detection [137,138]. Currently, more than 40 clinical trials of TEXs biomarkers are in progress, according to the Clinical Trials Registry website (clinicaltrials.gov).

#### 2.2.2. Microvesicles

Unlike exosomes, microvesicles are a type of Evs formed directly by the cytoplasmic membrane through outward outgrowth form [139]. The biogenesis of microvesicles involves local stiffness and curvature changes caused by the rearrangement of lipids and proteins on the cytoplasmic membrane, followed by the vertical transport of molecular material to the plasma membrane to form specific contents, and finally, the use of surface contraction mechanisms to squeeze leading to the outward blistering of the plasma membrane to produce vesicles [91,92]. Microvesicles not only randomly wrap the cell contents but also selectively incorporate proteins and nucleic acids, etc., into the vesicles. It has been shown that ARF6 is one of the microvesicle regulatory selective proteins [140], which binds to Ras-related small GTP and is involved in the activation and regulation of intracellular somatic recycling and extracellular peripheral actin remodeling [141]. EGFR, Akt, MT1-MMP and other complex kinase receptors, β1 integrin receptor, MHC-I, VAMP3 and others enter microvesicles via ARF6-regulated intracellular somatic recycling [142,143,144]. Meanwhile, AFR6 can translocate the ARF6-Exportin5 axis of miRNA into microvesicles [145].

Tumor-derived microvesicles (TMVs) can deliver biologically active components, including oncoproteins, oncogenes, soluble factors, chemokine receptors, specific enzymes and microRNAs [146,147,148]. TMVs help tumor cells evade by exposing FasL proteins, tumor necrosis factor (TNF)-associated regulatory ligands, etc., to induce apoptosis in CD8^+^ T cells’ immune response [149,150]. Breast cancer cells or glioma cells secrete glutamyltransferase microvesicles to promote tumorigenesis and improve the tumor microenvironment [100]. In breast cancer cells cultured in a hypoxic environment, Rab22a is co-localized with microvesicles by a situation that regulates microvesicle formation and material sorting [151]. VEGF released from tumor cells is an influential factor in promoting tumor angiogenesis [152]. TMVs were reported to carry CD147 to promote VEGF secretion and thus induce angiogenesis in ovarian cancer [153].

Nevertheless, Zhang et al. observed that miRNA-29a/c carried by TMV inhibited the growth of vascular cells by suppressing VEGF expression in gastric cancer cells [154]. Thus, miRNA-loaded microvesicles could be designed to suppress tumor growth by blocking blood vessel generation. Multidrug resistance mediated by the plasma membrane multidrug efflux transporter, P-glycoprotein (P-gp), often leads to tumor treatment failure [155]. In breast cancer, the transient receptor potential channel 5 (TRPC5) protein carried by TMV was demonstrated to regulate the expression of P-gp and promote tumor drug resistance [156]. Dong et al. found that the TRPC5 blocking antibody T5E3 reduced P-gp expression in human breast P-gp production in human epithelial cells and converted drug-resistant cells to non-drug-resistant cells [157]. MVs can also merge with other biological membranes to transport anti-tumor drugs into cells as a type of EVs. Therefore, TMVs with homologous targeting might be suitable vehicles for anti-tumor therapy [76]. Meanwhile, TMVs can serve as vehicles carrying biological signals or molecules that facilitate tumor cell invasion and metastasis and serve as biomarkers to predict cancer patient prognosis [158].

#### 2.2.3. Apoptotic Bodies

Apoptotic vesicles refer to the condensation of chromatin during apoptosis leading to blistering of the plasma membrane enclosing the cell contents into distinct membrane-enveloped vesicles, which prevent toxic components of dying cells from damaging the cells [159,160]. Apoptotic vesicles are the largest extracellular vesicles, approximately 500–2000 nm in size, and typically contain DNA fragments, histones and cytoplasmic organelle fragments [93]. The markers of apoptotic vesicles are usually considered to be Annexin V, thrombospondin and C3b [94,161,162]. In the normal state, apoptotic vesicles are cleared by phagocytosis through specific interactions between recognition receptors on and specific changes in the membrane composition of apoptotic cells [162,163,164]. Recent studies have indicated that apoptotic vesicles are implicated in tumor progression, metastasis and microenvironment formation. Apoptotic vesicles carry DNA fragments capable of transferring at the level of adjacent but different cell types. Apoptotic vesicles can transport tumor DNA from human C-MYC and H-RASV12-transfected rat fibroblasts to wild-type mouse fibroblasts, activating the full tumorigenic potential of wild-type cells [165]. DNA packaged into lymphoma-derived apoptotic vesicles is phagocytosed by surrounding fibroblasts, leading to the fusion of lymphoma-derived DNA into the genome of fibroblasts [166]. In addition, apoptotic vesicles can transfer proteins to phagocytic cells such as macrophages and DCs for immunomodulation [167,168,169]. For example, macrophages can phagocytose apoptotic vesicles containing auto-antigens, suggesting that auto-antigens can be transferred to specialized phagocytes, providing new ideas for tumor vaccine research [170].

## 3. Engineering Tumor Cell-Derived Membrane Vesicles in Cancer Treatment

### 3.1. Encapsulation

In 2011, Zhang took the lead in proposing a camouflage strategy, using erythrocyte membrane wrapping material to obtain camouflage to evade the clearance of nanocarriers by immune cells, thus deriving a biomimetic delivery platform for nanocarriers modified with cell membranes [171]. At the same time, the encapsulated nanocarriers also help provide physical support for membrane vesicles, ensuring that the functional components are embedded in the membrane function. A variety of materials, including mesoporous silicon, polymers, magnetic nanoparticles, MOFs, gold nanoparticles and upconversion nanoparticles, are encapsulated to function in membrane vesicles [72,171,172,173,174,175]. Kroll et al. used TCMVs to encapsulate PLGA nanoparticles loaded with immune adjuvant CpG and achieved significant therapeutic and preventive effects (Figure 2) [81]. The use of PLGA nanoparticles with small particle sizes is beneficial to lymph node drainage, and CpG can efficiently induce the maturation of APCs cells to break the immunosuppression of the tumor microenvironment. TCMVs that retain cell surface antigenic proteins are wrapped on the surface of nanoparticles to more realistically mimic the surface structure of tumor cells and induce anti-tumor immune responses more efficiently. Li et al. developed a novel cancer vaccine with Fe_3_O_4_ magnetic nanoclusters as the core and anti-CD205-modified TCMVs as the coat [79]. Vaccine homing in lymph nodes was achieved using Fe_3_O_4_ magnetism. Meanwhile, camouflaged TCMVs act as reservoirs for various antigens, enabling subsequent multi-antigen responses. In addition, the modified anti-CD205 directed more nanoparticles to CD8^+^ DCs, promoting antigen cross-presentation. These specific benefits lead to the considerable proliferation of T cells with clonal diversity and cytotoxic activity.

### 3.2. Surface Modification

The surface of cell-derived membrane vesicles is composed of polysaccharides, proteins, lipids, etc. They provide various surface properties that allow modification by foreign materials. For example, thiol and amine groups on the membrane surface can interact with maleimide-modified nanoparticles on the membrane surface via thiol and carboxylation reactions [80,176,177]. Jia et al. engineered an integrated glioma exosome-based diagnostic and therapeutic platform that carried curcumin (Cur) and superparamagnetic iron oxide nanoparticles (SPIONs) [178]. These TEXs were conjugated to peptides targeting neuropilin-1 by click chemistry for glioma-targeted exosome imaging and therapeutic function. Experiments with glioma cells and in-suit glioma models demonstrate that such engineered TEXs can smoothly cross the blood–brain barrier, providing favorable conditions for targeted imaging and treatment of gliomas. Chemical coupling does not destroy the integrity of membrane vesicles and has the advantages of high selectivity and fast reaction speed. However, conditions such as osmotic pressure and temperature must be strictly controlled during the reaction. Additionally, the residual solvents should be removed to avoid rupture or denaturation of membrane vesicles after the reaction [179]. The surface modification of membrane vesicles by physical means such as hydrophobic interaction, receptor–ligand binding and electrostatic interaction is also a commonly used method [103,180]. Membrane vesicles have lipid bilayers and negative charges, enabling hydrophobic and positively charged materials to adsorb on membrane vesicles stably. Physical strategies are readily achieved by direct co-incubation with cells or EVs within a specific temperature range. DSPE-PEG is a commonly used auxiliary phospholipid, which can bind to RBD, maleimide, folic acid and other ligands to enhance the targeting of materials [181,182,183]. Yang et al. used mannose-modified DSPE-PEG to insert into TCMVs to enhance antigen presentation by utilizing mannose receptors carried on the surface of APCs [175]. Nakase binds cationic lipids to the surface of exosomes via electrostatic interactions, facilitating the uptake of exosomes [184]. Although the physical strategy is easy to operate and has high safety, it still suffers from the difficulty in controlling the adhesion strength, and the resulting vesicles may dissociate due to in vivo shear forces.

### 3.3. Membrane Fusion

To enhance the effect of tumor cell-derived membrane vesicles, the researchers used different kinds of materials to fuse them. In addition to excellent biocompatibility, senescent or damaged red blood cells are eliminated by cells such as macrophages and DCs in the spleen, providing spleen targeting capabilities [69,185]. Wang et al. designed a pH-responsive copolymer micelle camouflaged by the erythrocyte and TCMVs hybrid membranes. In the acidic tumor microenvironment, the micelle exhibited a membrane escape effect, which could promote recognition and interaction with tumor-associated macrophages [70]. Han et al. obtained a personalized vaccine by fusing red blood cells with TCMVs by ultrasound and extrusion. The vaccine particles can be effectively delivered to the spleen and activate the T cell immune response [186]. The fusion vesicles of DCs-derived cell membranes and TCMVs retain the antigens of tumor cells and carry costimulatory substances such as MHC-I, MHC-II, CD80 and CD86, and retain the antigen presentation and T cell activation functions of DCs [60,61,62,187]. The macrophages are highly infiltrating and can induce better tumor treatment and recurrence prevention after fusion with TCMVs [64,188]. The vesicles obtained by fusion of multiple TCMVs carry multiple tumor-specific antigens, which can treat multiple tumors and prevent recurrence [73,189]. Years of research have found a large number of pathogen-associated molecular patterns (PAMPs) on the surface of bacterial membranes, such as lipopolysaccharides, mannose, lifters and lectins. These PAMPs can induce the recognition and binding of highly expressed pattern recognition receptors (PRRs) on the surface of innate immune cells, thereby activating innate and adaptive immunity and clearing tumors [2,190]. Chen et al. designed hybrid vesicles of attenuated Salmonella fused to TCMVs [66]. This hybrid vesicle inherits the immune functions of both parents and exhibits a robust immune response to tumor cells. In addition, loading indocyanine green (ICG) inside the hybrid vesicles can induce local photothermal effects, effectively destroying solid tumors and inhibiting tumor recurrence. Zuo et al. constructed hybrid vesicles fused with E. coli membranes and TCMVs to inhibit tumor metastasis (Figure 3) [191]. Due to the immunogenicity of bacterial membranes, hybrid vesicles are more likely to be accumulated in the draining lymph nodes than single TCMVs. The activated innate immune system further activates the adaptive immune response involving T lymphocytes, effectively inhibiting tumor growth, recurrence and metastasis to the lung. Furthermore, hybrid vesicles induced adaptive immune responses in a syngeneic bilateral tumor model, reversibly demonstrating the effect of individualized immunotherapy.

### 3.4. Genetic Engineering

Genetic engineering has higher controllability, safety and flexibility than membrane fusion technology. Genetic engineering transfers the genetic information of living cells into membrane vesicles through physical methods such as viral vectors, cationic polymers, electroporation and microinjection [53,54,192,193]. Rao et al. fused TCMVs containing SiRPα variant, M1 macrophage membrane and platelet membrane to obtain hybrid fusion vesicles. By blocking the CD47/SIRP innate immune pathway and promoting M2 to M1 repolarization in the tumor microenvironment, local recurrence and metastasis of malignant melanoma were significantly prevented [194]. Meng et al. constructed a fusion cell vesicle of a high-affinity SIRPα variant and PD-1 (Figure 4) [73]. First, 4T1 cells and B16F10 cells could express SIRPα and PD-1, respectively, by gene editing, and the cell membranes of the two were fused to obtain fused TCMVs. Simultaneously blocking the CD47/SIRP and PD-1/PD-L1 immunosuppressive axis via SIPRα and PD-1 promotes antigen presentation by macrophages and DCs and enhances anti-tumor T cell immunity.

Similarly, taking advantage of the targeting properties of TEVs, introducing genetic information such as small interfering RNA (siRNA) and miRNA from other cells into TEVs can induce the expression of transgenic proteins in target cells and prevent RNA from being degraded by RNases [194,195,196,197]. Ohno targeted miRNA delivery to EGFR-expressing cancer tissues using modified exosomes with GE11 peptide or EGF on the surface [198]. Morishita et al. genetically engineered melanoma cells by transfection with a plasmid encoding a streptavidin-lacadherin fusion protein to generate genetically engineered exosomes. A vaccine combining these engineered exosomes with a biotinylated CpG adjuvant in vitro induced strong anti-tumor effects in a mouse model of melanoma [112]. High mobility group nucleosome-binding protein 1 (HMGN1) is a protein adjuvant that can enhance the response of DCs to exogenous antigens and continuously induce Th1 immune responses [199]. Zuo et al. directly anchored the functional N-terminal domain of HMGN1 (N1ND) to TEXs through CP05, an exosome anchoring peptide, which enhanced the maturation and activation of DCs and accelerated the generation of memory T cells. The approach generated strong and durable anti-tumor immunity [107]. Shi et al. found that genetically engineered exosomes with anti-CD3 and anti-HER2 antibodies led SMART-Exos to efficiently and selectively induce HER2-expressing tumor-specific immunity, thereby providing a new tumor immunotherapy idea for HER2-positive breast cancer [200,201].

## 4. The Application of Tumor Cell-Derived Membrane Vesicles to Personalized Immunotherapy

Personalized therapy is the most suitable therapy for the patient, taking into account the individual circumstances of the patient’s genetic information, epigenetic and environmental factors. It also ensures the effectiveness of prescriptions, reduces the adverse reactions caused by traditional treatments and costs more time for treatment [202]. The development of tumor cell-derived membrane vesicles technology provides precise targeting of tumor tissue for personalized immunotherapy. In addition, tumor cell-derived membrane vesicles can combine with stimuli-responsive treatments such as light, sound, magnetism, pH and reactive oxygen species to improve tumor personalized treatment effects, promote tumor regression and inhibit metastasis and recurrence [74,203,204].

### 4.1. Cancer Vaccines

As emerging tumor immunotherapy, tumor vaccines utilize the administration of tumor cell-associated antigens and other immune stimulatory signals to train the body’s immune system to recognize and fight tumors [65,205,206]. It possesses the advantages of high specificity, low cost and few adverse reactions. Corresponding tumor vaccines such as breast and bladder cancer have also entered the clinical trial stage. Considerable inherited heterogeneity exists between patients with the same type of cancer [71,207]. The complexity and diversity of antigens result in significant differences in antigen presentation between individuals with the same tumor type, leading to various efficacy between different populations. Vaccines from lysates of tumor tissues or tumor cell lines contain intact tumor cell-specific antigens.

Tumor cell-derived membrane vesicles-coated drugs are a potent anticancer weapon. In addition to inducing a highly specific immune response, they also can target homologous tumor cells [81,110]. Ye et al. used melanoma tumor cell lysate as antigen combined with photothermal therapy to enhance antigen uptake by DCs, promote T cell migration and local pro-inflammatory cytokine production and effectively target primary and distal secondary primary tumors [208]. Yang et al. proposed to coat imiquimod adjuvant nanoparticles with mannose-modified TCMVs, so that the vaccine can be taken up and presented by antigen-presenting cells in large quantities, triggering an anti-tumor immune response [175]. Ma et al. prepared a tumor cell lysate-based nanovaccine, which induced tumor-specific T cell responses, and both adaptive and innate immune responses against cancer cells were activated by the nanovaccine [209]. DCs are important immune cells linking innate and adaptive immunity and serve as vaccine targets. Lu et al. reported a hydrogel personalized vaccine loaded with granulocyte-macrophage colony-stimulating factor (GM-CSF) using surgically resected tumor cell lysates as antigens. DCs are recruited by releasing GM-CSF from the hydrogel, thereby providing a fully personalized tumor antigen repertoire, exhibiting excellent tumor-suppressive effects in postoperative tumor models [210].

However, the complex intracellular proteins and organelle components in whole-cell lysates can disrupt the recognition and presentation of antigens [211]. In contrast to whole cell lysates, TCMVs have complete cell membrane protein antigens (including neoantigens) and do not contain complex cytoplasmic proteins and components, making them the best tumor vaccine candidates. TCMVs vaccine triggers individualized immune responses against corresponding tumors and induces more effective and durable anti-tumor immune responses while avoiding tumor cytoplasmic protein interference and immune escape caused by downregulation of partial antigen expression [67,212]. Kroll et al. designed a nanovaccine with TCMVs encapsulating CpG, using the antigenic proteins on TCMVs to activate the body-specific immune response, leading to an anti-tumor effect [81]. Xiong et al. reported a calmodulin-expressing TCMVs wrapper-loaded R837 nanovaccine that used calmodulin exposed on the surface of TCMVs to induce active uptake of the vaccine by DCs, enabling simultaneous delivery of adjuvant and antigen and departure of a personalized immune response against 4T1 tumors [67]. Xu et al. constructed a fluorinated polymer-based personalized tumor vaccine that showed a robust immune memory effect in several postoperative models and effectively protected against tumor recurrence [213]. Zhang et al. fused the Toll-like receptor agonist monophosphoryl lipid A (MPLA) into the phospholipid bilayer of TCMVs to increase the activity of APCs, thereby activating CD8^+^ T cells to kill tumors [86]. Liu et al. prepared a DCs and TCMVs fusion vesicle tumor vaccine (Figure 5) [62]. The fusion of the two immune-associated cells resulted in high expression of tumor antigen complexes and immune costimulatory molecules on the fusion vesicles, allowing the vaccine particles to exert APCs to stimulate T cell immune activation and induce a favorable immune response. In addition, eukaryotic-prokaryotic nanovaccines constructed using TCMVs with bacterial extracellular vesicles showed powerful therapeutic and therapeutic effects, providing new perspectives on the design of tumor immunogenicity, adjuvant and scalable vaccine platforms [66].

The immunogenicity of TEXs could be improved by gene editing [214]. TEXs loaded with IL-2 and IL-18 cytokine genes could induce potent specific killing responses [215,216]. Yin et al. anchored the TLR4 agonist HMGN1 short functional peptide to TEXs from different sources through the exosome anchoring peptide CP05 and obtained a more immunogenic DC vaccine, especially in the middle and late stages with low immune response rate. In the mouse model of hepatocellular carcinoma, the tumor microenvironment was significantly improved, and long-term immune response and tumor inhibition were obtained [107]. ICD can enhance tumor antigen exposure, promote the release of immune-stimulating tumor cell content, and facilitate the uptake of dying tumor cells by DCs. Huang et al. constructed a combination of TLR3 agonist Hiltonol and ICD inducer ELANE to engineer TEXs that can activate DCs in situ in a mouse xenograft model of poorly immunogenic triple-negative breast cancer and tumor organs derived from patients with hot tears; both produced effective tumor suppression [104]. In conclusion, engineered tumor-cell-derived membrane vesicles have a promising application in developing anti-tumor vaccines due to their great advantage of carrying tumor-specific antigens.

### 4.2. Immune Checkpoint Therapy

Immune checkpoint therapy (ICT) has achieved therapeutic effects in a variety of tumor models, but in clinical trials, the response rates of different patients to ICT vary widely, and most patients cannot benefit from ICT therapy. The development of tumor cell-derived membrane vesicles technology provides a new idea for ICT. Li et al. designed an injectable hydrogel based on TCMVs with the powerful ability to simultaneously reprogram local tumors and circulating exosomal PD-L1 to facilitate PD-L1-based immune checkpoint therapy (Figure 6) [74]. Using sodium oxidized alginate-modified TCMVs as a gelling agent, hydrogels were formed in vivo with Ca^2+^ channel inhibitors (DMA) and cell cycle protein-dependent kinase 5 (Cdk5) inhibitors to create an immune ecology as an antigen pool. Reducing the amount of circulating exosomal PD-L1, decreasing genetic PD-L1 expression in tumor cells, attenuating IFN-γ-induced PD-L1 adaptive immune tolerance and achieving downregulation of PD-L1 expression in tumor cells and exosomes.

miRNA-424 in TEVs inhibits the CD80/86-CD28 costimulatory pathway in DCs and T cells, resulting in resistance to ICT. TEVs with knockdown of miRNA-424 increase the efficacy of immune checkpoint blockade therapy and stimulate anti-tumor immunity [105]. CD47 is overexpressed on a variety of tumor cells and activates “don’t eat me” signaling by binding to SIRPα, causing immune escape of tumor cells from the mononuclear phagocyte system. Researchers designed exosomes containing SIRPα variants as immune checkpoint blockers, thereby antagonizing the interaction between CD47 and SIRPα, inducing enhanced tumor phagocytosis, and triggering an effective anti-tumor T-cell response [53,103,106].

### 4.3. Combination Therapy

In order to achieve complementary advantages and functional amplification of various therapeutic methods, researchers have conducted many studies on the combination of immunotherapy with other methods such as chemotherapy, photothermal and photodynamic therapy [217,218,219,220]. Wu et al. developed a chemoimmunotherapy-based TCMVs nanoparticle, using the homotypic targeting ability of TCMVs to deliver DOX to the tumor site, causing tumor cell death to form a tumor in situ vaccine [78]. TEVS-loaded chemotherapy drugs can promote drug uptake and reverse drug resistance in tumor-regenerative or stem-like cancer cells [221,222,223]. Remarkable achievements have been made in photodynamic and photothermal tumor therapy by exploiting the homologous targeting properties of TEVs [77,108,111,224]. Wang used TCMVs to encapsulate ovalbumin-assembled nanoparticles encapsulating the photosensitizer Ce6 [75]. The ROS generated after illumination significantly enhances the cross-presentation efficiency of antigens, which can effectively initiate immune cascade reactions and improve the effectiveness of traditional photodynamic therapy. Zheng et al. fed cells with attached CpG gold nanorods and then produced apoptotic bodies loaded with gold nanorods by UV light irradiation of cells [109]. Upon near-infrared light irradiation, the photothermal effect triggered by the gold nanorods effectively eliminated the tumor.

At the same time, the strategy relied on co-stimulation with immune agonist, CpG, and tumor-associated antigens released by photothermal therapy in situ, promising to elicit a practical, durable tumor-specific immune response advantageous for alleviating immunosuppression. Zhen et al. constructed a biomimetic nanoparticle (CS-I/J@CM NPs) combining light, sound and immune checkpoint therapy for the treatment of glioblastoma (Figure 7) [63]. The encapsulation of TCMVs can improve the targeting ability and enrichment of particles at tumor sites. Cu2-xSe particles possess the properties of light-responsive Fenton-like catalytic induction of immunogenic cell death and alleviation of tumor hypoxia, repolarizing M2 macrophages to M1 type, thereby alleviating the immunosuppressive microenvironment of glioblastoma. The release of indoximod effectively blocked IDO-induced Treg cell infiltration in tumor sites. JQ1 can reduce the expression of PD-L1 on cancer cells. Under NIR II light irradiation, nanoparticles transformed glioblastoma from cold tumor to hot tumor, and the CD8^+^ T cells in the tumor increased significantly, showing a good therapeutic effect.

## 5. Concluding Remarks and Future Perspectives

From synthetic materials such as functionalized liposomes and micelles inspired by biomembranes to tumor cell-derived membrane vesicle-coated nanoparticles that bind to cell membranes, researchers have pursued suitable carrier materials for tumor immunotherapy. This review focuses on TDMV technology and its application in personalized immunotherapy. Tumor immunotherapy has yielded exciting results in several clinical trials. At the same time, achieving precise targeting and universal efficacy is also a problem for tumor immunotherapies in clinical trials. The choice of any TDMVs can target homologous tumors, overcoming the disadvantages of synthetic particles. TDMVs exploit the biological properties of cells, such as the ability to cross biological barriers, circulate in the body for long periods, interact with other cells and the capability to pass through biological barriers. In particular, the ability to pass through biological barriers and modify tissue toxicity effectively protects the drug from its effects. Unique approaches are utilized by introducing integrated biological components and functions that have synergistic effects and enhance the performance of TDMVs. For example, ligands composed of antibodies, peptides and proteins can be integrated into cell membranes to improve the function of cancer immunotherapy. TDMVs carry tumor cell proteins or genetic material, and their modification enables personalized therapy for all patients, especially for the development of tumor vaccines.

Cancer immunotherapy based on TDMVs technology is an attractive option due to the excellent biocompatibility and versatility, but in practice, there are many challenges to overcome to achieve widespread clinical application. First, the mechanism of action of TDMVs with tumor cells has not yet been fully explained. For example, the biological information carried by TEVs is stochastic in nature, and the mechanism of carrying is still unclear to achieve effective and precise regulation. Its mode of communication with the cell remains unclear. Second, whether the batch reproducibility, homogeneity and storage stability of TDMVs meet the quality manufacturing specification (GMP) standards also limit their further clinical translation. Third, biosafety issues are owing to unknown biological mechanisms. The current system for evaluating biosafety in the clinical setting is not robust, making the safety assessment of TDMVs unconvincing. In addition, personalized immunotherapies based on TDMVs currently have long lead times and high costs, and further exploration of optimization options is needed. However, with greater understanding and creative ideas, the field of TDMV -based nanotechnology can certainly circumvent these issues and drive cell membrane nanotechnology closer to accurate clinical applications.

## Figures and Tables

**Figure 1 pharmaceuticals-15-00876-f001:**
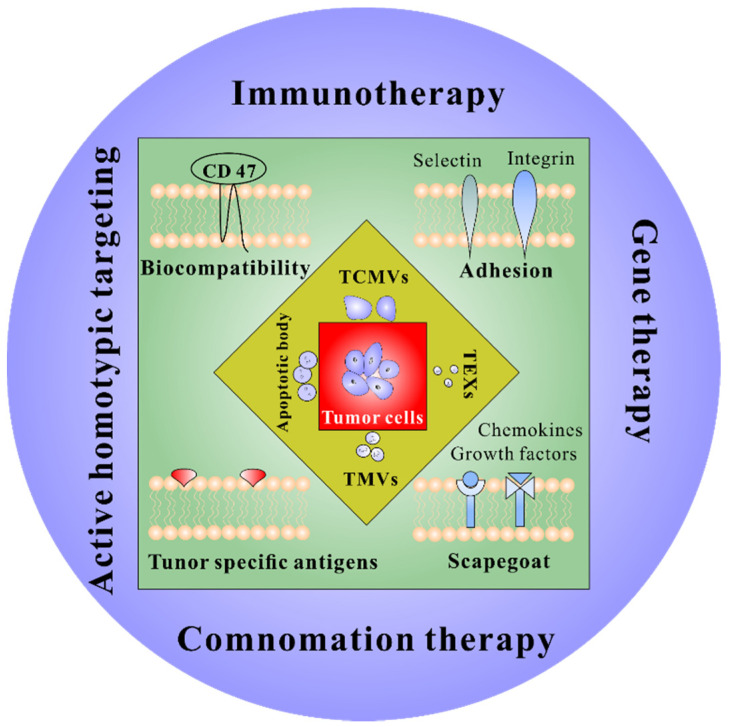
Properties and application prospects of TDMVs. TCMVs, tumor-derived cell membrane vesicles; TEXs, tumor-derived exosomes; TMVs, tumor-derived microvesicles.

**Figure 2 pharmaceuticals-15-00876-f002:**
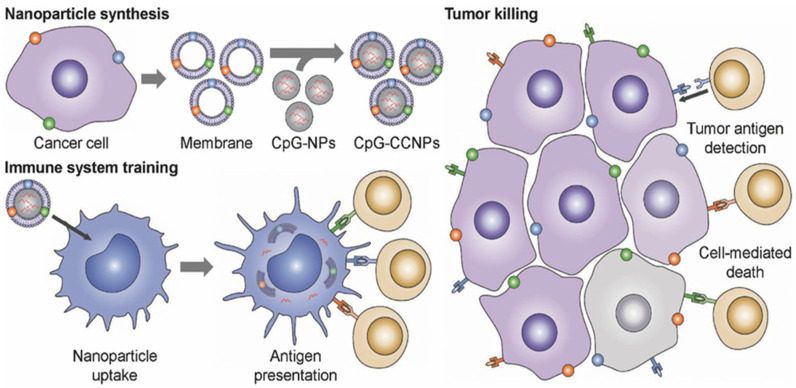
Schematic diagram of TCMVs-based CpG-anti-tumor vaccines. Use of cancer cell membranes carrying tumor-specific antigens to wrap CpG-loaded nanoparticles to generate nanoparticle tumor vaccines. The tumor-specific antigens on the surface of TCMVs promote uptake and presentation by antigen-presenting cells, activating multiple specific T cells that act to monitor and kill tumor cells. Reproduced with permission [81]. Copyright 2017, Wiley-VCH.

**Figure 3 pharmaceuticals-15-00876-f003:**
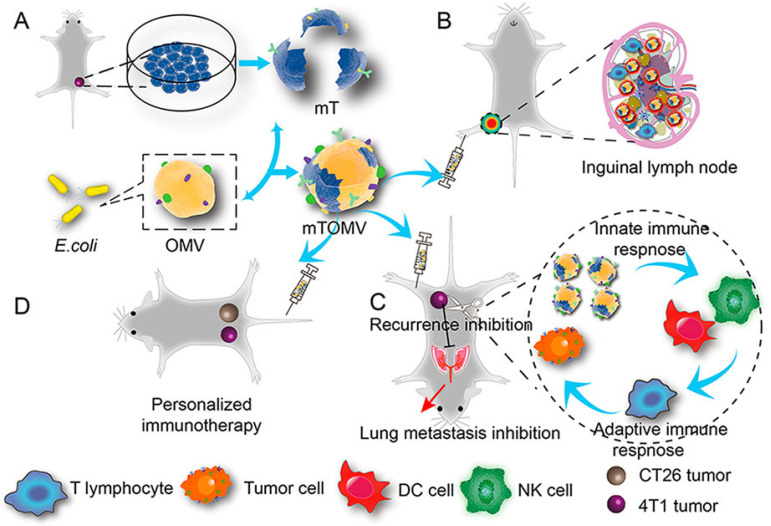
Schematic illustration of fusion vesicles of bacterial outer membrane and TCMVs for personalized immunotherapy. (**A**) The fabrication of fusion vesicles. (**B**) Accumulation and retention behavior of inguinal lymph nodes following right posterior intraplantar injection. (**C**) Fusion vesicles inhibit tumor lung metastasis. (**D**) Bilateral tumor model to validate the effect of fusion vesicle personalized immunotherapy. Reproduced with permission [191]. Copyright 2021, American Chemical Society.

**Figure 4 pharmaceuticals-15-00876-f004:**
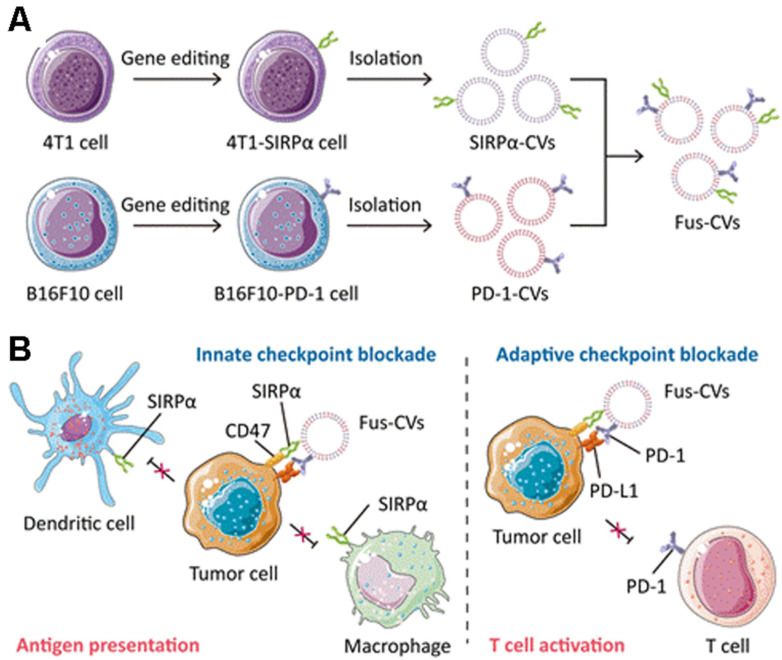
Gene-edited fusion vesicles for multi-targeted immune checkpoint therapy. (**A**) SIRPα variants CV1 and PD-1 were overexpressed on 4T1 and B16F10 cancer cells, respectively, and fusion vesicles were then prepared. (**B**) Fusion vesicles promote antigen uptake and presentation by antigen-presenting cells and enhance anti-tumor T-cell immunity by blocking CD47/SIRPα and PD-1/PD-L1immunosuppressive axis. Reproduced with permission [73]. Copyright 2021, Wiley-VCH.

**Figure 5 pharmaceuticals-15-00876-f005:**
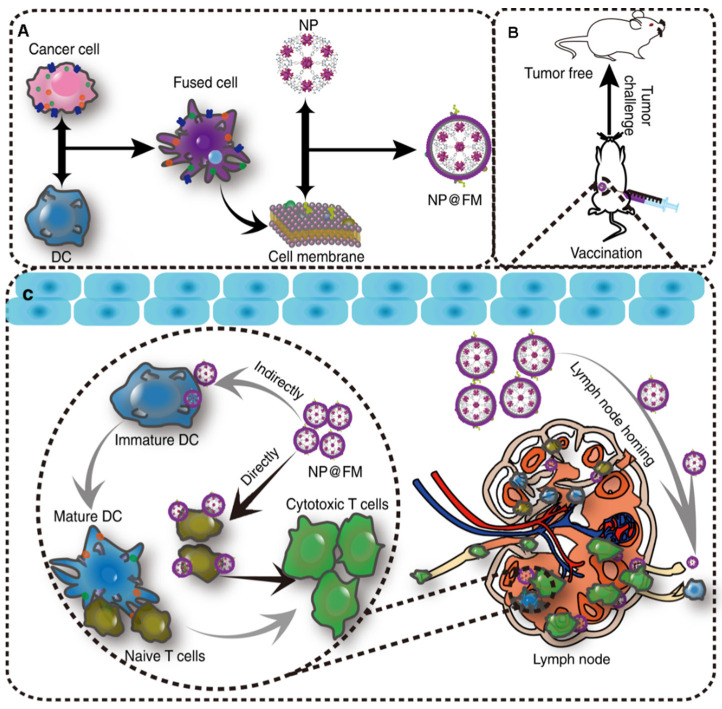
Fusion vesicles encapsulating MOFs for tumor prevention. (**A**) Preparation of fused vesicles encapsulating MOFs. (**B**) Fusion vesicle vaccination for tumor prophylaxis. (**C**) Mechanisms by which fusion vesicles induce immune response. Reprinted/adapted with permission from Ref. [62]. 2019, Springer Nature.

**Figure 6 pharmaceuticals-15-00876-f006:**
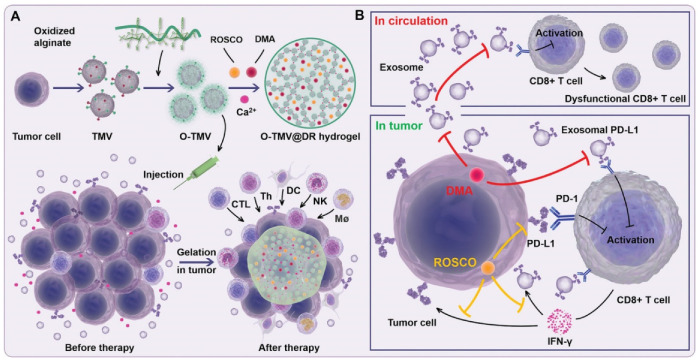
Schematic diagram of TCMVs-based injectable hydrogels for ICT. (**A**) Oxidized sodium alginate is adsorbed on the surface of TCMVs to form a gel, and after injection into the tumor, Ca^2+^ in the microenvironment would chelate with the particles to form a gel, causing an antigen reservoir effect and continuous recruitment to activate antigen-presenting cells and lymphocytes. (**B**) Adding DMA and ROSCO to the gel blocks Ca^2+^ entry into tumor cells and inhibits the secretion of circulating PD-L1 and tumor PD-L1 exosomes. Reproduced with permission [74]. Copyright 2021, Wiley-VCH.

**Figure 7 pharmaceuticals-15-00876-f007:**
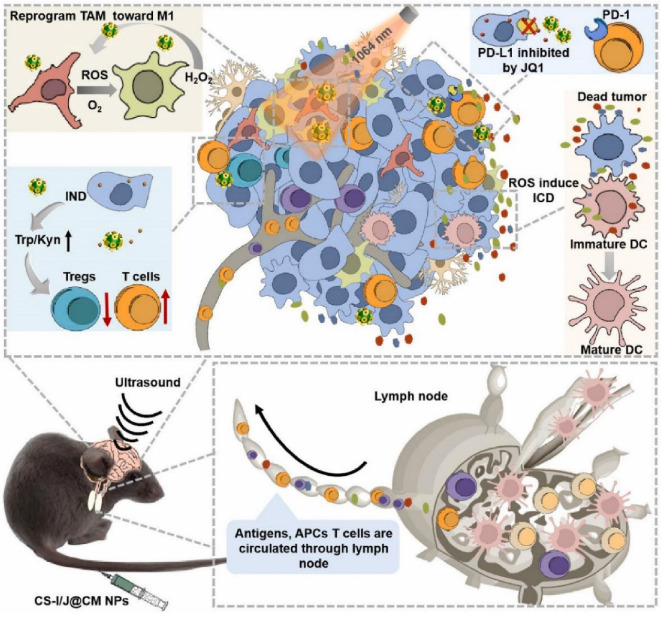
Integrated CS-I/J@CM NPs remodel the tumor immunosuppressive microenvironment to improve glioblastoma immunotherapy [63].

**Table 1 pharmaceuticals-15-00876-t001:** Representative applications of TCMVS in tumor therapy.

TCMVs Sourse	Engineering Strategy	Drug	Application	Ref.
Ovarian cancer cell	DC/TCMVs Fusion; PLGA core	CpG ODN	Vaccine	[60]
4T1 cell	DC/TCMVs fusion; MOF core	photosensitizer	PDT and vaccine	[61]
4T1 cell	DC/TCMVs fusion; MOF core	-	vaccine	[62]
Glioblastoma cell	Cu_2_-xSe NPs core	PD-L1 inhibitor and indoximod	ICT, PDT and ICD	[63]
Osteosarcoma cell	Macrophage cell membrane/TCMVs fusion; PLGA core	PTX	targeted tumor	[64]
solid tumor	Bacterial membrane/TCMVs fusion	-	vaccine	[64,65]
B16F10 cell	Bacterial membrane/TCMVs fusion; PLGA core	ICG	Vaccine and PTT	[66]
B16F10	RBC/TCMVs fusion; CuS NPs core	DOX	Chemo-immunotherapy	[34]
4T1 cell	PLGA core	R837	vaccine	[67,68]
Ovarian cancer cell	RBC/TCMVs fusion; Fe_3_O_4_ core	ICG	PTT and Immunotherapy	[69]
4T1 cell	RBC/TCMVs fusion; Fe_3_O_4_ core	CSF-1R inhibitor:	Immunotherapy	[70]
HepG2 cell	Prussian blue Nps core	-	PTT	[56]
4T1 cell	alginate gel encapsulation	anti-PD-1 antibodies	vaccine	[71]
4T1 cell	PAMAM core	DOX	targeting and anti-metastasis treatment	[72]
HCT116	F127 core	R837	vaccine	[33]
4T1 cell and B16F10 cell	Encoded SIRPα and PD-1	-	ICT and CD47 blockade	[73]
B16F10	-	DMA and Cdk5 inhibitor	ICT	[74]
Hela cell	PLGA core	PTX and siRNA	Chemo-immunotherapy	[48]
B16-OVA cell	OVA assembly core	Ce6	PDT	[75]
lung carcinoma cell	-	Dox and Sorafenib	ICD and ICT	[49]
4T1 cell	Surface-anchored CD80 and IL-12	-	vaccine	[76]
MCF-7	Au NPs core	MicroRNA	cancer diagnosis	[52]
4T1 cell and B16F10 cell	thermosensitive hydrogel encapsulation	black phosphorus	Vaccine, PTT and ICT	[77]
B16F10 cell	cationic polymers core	DOX	Chemo-immunotherapy	[78]
Tumor cell	Surface-anchored anti CD205	-	vaccine	[79]
B16-OVA	Surface-anchored mannose; PLGA core	R837	vaccine	[80]
B16F10	PLGA core	CpG	vaccine	[81]

ICD, Immunogenic cell death; ICT, immune checkpoint therapy; NPs, nanoparticles; PDT, photodynamic therapy; PTT, photothermal therapy.

**Table 2 pharmaceuticals-15-00876-t002:** Classification of different types of extracellular vesicles.

	Exosomes	Microvesicles	Apoptotic Bodies
Size	20–100 nm	50–1000 nm	500–2000 nm
Biogenesis	The inner membrane forms multivesicles within the cell that fuses with the plasma membrane to release exosomes into the extracellular compartment	Local changes in plasma membrane stiffness and curvature, cell surface shrinkage and outward blistering	Released by belting of apoptotic cell membrane
Contents	mRNA, microRNA, cytoplasmic and membrane protein, MHC	mRNA, miroRNA, noncoding RNAs, cytoplasmic and membrane protein	nuclear fractions, cytoplasmic protein, cell organelles
Biomakers	Tetraspanins, ESCRT peoteins, flotillin, TSG101	Integrins, seltctins, CD40 ligand	Phosphatidylserine, annexin V
Ref.	[89,90]	[91,92]	[93,94]

**Table 3 pharmaceuticals-15-00876-t003:** Representative applications of TEVs in tumor therapy.

TEVs Sourse	Engineering Strategy	Drug	Application	Ref.
MDA-MB-231 cells	alpha-lactalbumin-engineered TEVs	ELANE and Hiltonol	Vaccine and ICD	[104]
Colorectal cancer cell	-	microRNA 424	immunotherapy	[105]
CT26 cell	thermosensitive liposomes/gene-engineered TEVs Fusion	-	PTT and CD47 blockade	[106]
HEPA1–6	-	HMGN1	vaccine	[107]
4T1 cell	AIE luminogen/TEVs fusion	Dexamethasone	PDT	[108]
EL4 cell	gold-silver nanorods core	CpG	PTT and immunotherapy	[109]
HER2 expressing breast cancer	encoded anti-CD3 and anti-HER2 antibodies	-	immunotherapy	[110]
Serum exosomes from tumor-bearing mice	-	black phosphorus	PTT and vaccine	[111]
B16BL6 cells	encoding streptavidin lactadherin protein	CpG	immunotherapy	[112]

## Data Availability

Data sharing not applicable.

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
