# Peer review of "Tumor-Derived Membrane Vesicles: A Promising Tool for Personalized Immunotherapy"

_pharmaceuticals, 2022, doi:10.3390/ph15070876_

Round 1

Reviewer 1 Report

In this manuscript, the authors summarize current knowledge regarding the use of tumor-derived membrane vesicles for personalized immunotherapy. The info presented is sufficiently exhaustive and clearly presented. Overall, we recommended therefore publication after minor revision.

Following points should be addressed:

-          English language should be reviewed. Few passages are not clear or incorrect (e.g: line 18: “..make it widely..” should be “..make them widely..”; line 19-21 “In particular, …” needs revision; line 558 “the immunogenicity…by gene editing” needs revision (e.g.”…could be improved by gene editing”).

-          Line 49-50: Microbial community is crucial in cancers of GI tract but have less impact on other cancer types. I suggest the authors should focus here on the immunosuppressive aspects of the TME in solid tumors which is a more general and well established mechanism linked to cancer immune escape and immunotherapy failure.

-          Line 73-5: reference missing

-          Line 99-100: I believe this sentence is redundant and irrelevant here. I suggest to remove it

-          Use of acronyms should be revised, sometimes the same acronym is introduced twice e.g. TCMV lines 94 and 145), sometimes they are not always used e.g. TCMVs line 501

-          Line 293: “microvesicles” is mispelt

-          Figure legend 2 is misplaced in the text

-          Line 518: “they can also target homotypic”: not clear to me what the authors mean

-          Figure 3: “response” is misspelt in the figure

Author Response

Response to Reviewer 1 Comments

Point 1: English language should be reviewed. Few passages are not clear or incorrect (e.g: line 18: “..make it widely..” should be “..make them widely..”; line 19-21 “In particular, …” needs revision; line 558 “the immunogenicity…by gene editing” needs revision (e.g.”…could be improved by gene editing”).

Response 1: Thank the reviewer for the question about the grammatical issues, we have revised the grammatical issues in the manuscript, and rewrite Lines 19-21 to “In particular, personalized vectors prepared using the editable properties of cells can help in the design of personalized vaccines.”.

Point 2: Line 49-50: Microbial community is crucial in cancers of GI tract but have less impact on other cancer types. I suggest the authors should focus here on the immunosuppressive aspects of the TME in solid tumors which is a more general and well established mechanism linked to cancer immune escape and immunotherapy failure.

Response 2: Thank the reviewer for pointing this out, we modified Lines 49-50 in the manuscript to “The gap between the tumor microenvironment in the body is the main reason for the significant gap in the efficacy of tumor immunotherapy. In particular, the immunosup-pressive microenvironment and high interstitial pressure in solid tumors make it difficult for drugs to penetrate and act inside the tumor, thus contributing to the immune escape of tumor cells and making treatment very difficult.”.

Point 3: Line 73-5: reference missing

Response 3: Thank the reviewer for the question about the reference, we added the ref. 21 (DOI 10.1016/j.jconrel.2021.05.001), which reports accelerated blood clearance from PEG.

Point 4: Line 99-100: I believe this sentence is redundant and irrelevant here. I suggest to remove it

Response 4: Thank the reviewer for pointing this out, we have removed it at Line 99-100.

Point 5: Use of acronyms should be revised, sometimes the same acronym is introduced twice e.g. TCMV lines 94 and 145), sometimes they are not always used e.g. TCMVs line 501.

Response 5: Thank the reviewer for pointing this out, we have corrected to explain the abbreviations only when they first appear in the manuscript.

Point 6: Line 293: “microvesicles” is mispelt

Response 6: Thanks to the reviewer for the suggestion, the term "microvesicles" is a term used to refer to a type of extracellular membrane vesicle. Therefore we don’t  make changes in the manuscript.

Point 7: Figure legend 2 is misplaced in the text

Response 7: Thank the reviewer for pointing this out, we have placed Figure 2 caption below the figure.

Point 8: Line 518: “they can also target homotypic”: not clear to me what the authors mean

Response 8: Thanks to the reviewer for the suggestion, we modified Lines 518  in the manuscript to “In addition to inducing a highly specific immune response, they also can target homolo-gous tumor cells”. Sorry for the trouble caused by the unclear marking.

Point 9: Figure 3: “response” is misspelt in the figure

Response 9: Thank the reviewer for pointing this out, we have corrected in the manuscript.

Reviewer 2 Report

This work is generally nicely written, well-thought, and well-organized. It comprises a comprehensive review of the topic area reflected in the heading, i.e., characterization of tumor-derived membrane vesicles and their application in personalized immunotherapy. After a brief introduction to the problems of treatment of cancer diseases, the Authors discuss in detail the unique advantages of tumor cell-derived membranes, exosomes, microvesicles, and apoptotic bodies indicating their benefits and limitations. These considerations are closely related to the results of experimental studies that I consider the strong point of the work. This review also provides valuable and convincing information regarding the need to develop engineering tumor cell-derived membrane vesicles in cancer treatment. In my opinion, the manuscript can be interesting for the readers of Pharmaceuticals and should be published after making a few corrections.

Some minor comments:

·        At line 18: ‘proteins properties’? Is ‘properties’ needed here?

·        At lines 29 – 30: Maybe it would sound better: ‘According to the International Agency for Research on Cancer (IARC), 19.3 million new cases and million cancer deaths were estimated worldwide in 2020’.

·        At lines 33 – 36: The sentence starting with ‘Tumor…’ should be change to sound more clear.

·        Figure 1 caption place under the figure.

·        Chose: DOX (at line 110) or Dox (at line 152), anti-tumor or antitumor?

·        At line 141: double ‘each other’.

·        Does adriamycin (at lines 150, 253) have to be in capital?

·        At line 196: Delete ‘them’.

·        At line 202: Explain ‘TEV’ at the first use.

·        At line 202: Should be ‘interacting” instead of ‘interact’.

·        At line 294: ‘Evs’? What does it mean?

·        At line 308: Isn't it about TEXs (see line 115)? An explanation of the abbreviation is no needed here.

·        Figure 2 caption: Change the font and place it under the Figure.

·        At lines 398 – 399: Double ‘integrity’.

·        Figures 4 and 5: Save A, B, C in capitals as in other figures and figure captions.

·        At line 501: tumor

·        At lines 632 – 639: Lack of reference(s).

There are also double or missing spaces in some places that should be corrected.

Author Response

Response to Reviewer 2 Comments

Point 1: At line 18: ‘proteins properties’? Is ‘properties’ needed here?

Response 1: Thank the reviewer for the question about the grammatical issues, we have removed “properties”  from the manuscript.

Point 2: At lines 29 – 30: Maybe it would sound better: ‘According to the International Agency for Research on Cancer (IARC), 19.3 million new cases and million cancer deaths were estimated worldwide in 2020’.

Response 2: Thank the reviewer for the suggested language changes, which made the article more professional. We have changed it in the manuscript to " According to the International Agency for Research on Cancer (IARC), 19.3 million new cases and million cancer deaths were estimated worldwide in 2020".

Point 3: At lines 33 – 36: The sentence starting with ‘Tumor…’ should be change to sound more clear.

Response 3: As suggested by the reviewer, we have changed lines 33-36 in the manuscript to " In contrast to conventional treatments, tumor immunotherapy, which refers to the regula-tion of the patient's immune system to fight against tumors, has become the mainstream of research and clinical practice".

Point 4: Figure 1 caption place under the figure.

Response 4: Thank the reviewer for pointing this out, we have placed Figure 1 caption below the figure.

Point 5: Chose: DOX (at line 110) or Dox (at line 152), anti-tumor or antitumor?

Response 5: Thank the reviewer for the suggested language changes, we have modified the entire manuscript to “DOX” and “anti-tumor”.

Point 6: At line 141: double ‘each other’.

Response 6: Thank the reviewer for pointing this out, we have corrected line 141.

Point 7: Does adriamycin (at lines 150, 253) have to be in capital?

Response 7: Thank the reviewer for the suggested language changes, we have lower-cased the first letter of adriamycin.

Point 8: At line 196: Delete ‘them’.

Response 8: Thank the reviewer for pointing this out, we have corrected in the manuscript.

Point 9: At line 202: Explain ‘TEV’ at the first use.

Response 9: Thank the reviewer for pointing this out, TEV refers to tumor extracellular vesicles, which we explained when TEV first appeared in the manuscript (line 103).

Point 10: At line 202: Should be ‘interacting” instead of ‘interact’.

Response 10: Thank the reviewer for pointing this out, we have corrected in the manuscript.

Point 11: At line 294: ‘Evs’? What does it mean?

Response 11: Thank the reviewer for pointing this out, EVs refers to extracellular vesicles, which we explained when TEV first appeared in the manuscript (line 195).

Point 12: At line 308: Isn't it about TEXs (see line 115)? An explanation of the abbreviation is no needed here.

Response 12: Thanks to the reviewer's suggestion, we have removed the description of TEXs in line 294.

Point 13: Figure 2 caption: Change the font and place it under the Figure.

Response 13: Thank the reviewer for pointing this out, we have placed Figure 2 caption below the figure.

Point 14: At lines 398 – 399: Double ‘integrity’.

Response 14: Thank the reviewer for the suggested language changes, we have corrected in the manuscript.

Point 15: Figures 4 and 5: Save A, B, C in capitals as in other figures and figure captions.

Response 15: As suggested by the reviewer, we have corrected A, B, and C in Figure 4 and 5.

Point 16: At line 501: tumor

Response 16: Thank the reviewer for the suggested language changes, we have corrcted it to “improve tumor personalized treatment effects”.

Point 17: At lines 632 – 639: Lack of reference(s).

Response 17: Thanks to the reviewer for the suggestion, lines 632-639 in the manuscript are the description of Figure 7. Sorry for the trouble caused by the unclear marking.